# Full-Length Transcriptome Sequencing-Based Analysis of *Pinus sylvestris* var. *mongolica* in Response to *Sirex noctilio* Venom

**DOI:** 10.3390/insects13040338

**Published:** 2022-03-30

**Authors:** Chenglong Gao, Lili Ren, Ming Wang, Zhengtong Wang, Ningning Fu, Huiying Wang, Juan Shi

**Affiliations:** 1Beijing Key Laboratory for Forest Pest Control, Beijing Forestry University, Beijing 100083, China; gaocl0907@bjfu.edu.cn (C.G.); lily_ren@bjfu.edu.cn (L.R.); mingming66@bjfu.edu.cn (M.W.); wangzhengtong@bjfu.edu.cn (Z.W.); ning_fu@bjfu.edu.cn (N.F.); 2Sino-France Joint Laboratory for Invasive Forest Pests in Eurasia, INRAE-Beijing Forestry University, Beijing 100083, China; 3Jilin Forestry Survey and Design Institute, Changchun 130000, China; wanghuiying2677@163.com

**Keywords:** *Pinus sylvestris* var. *mongolica*, full-length transcriptome, *Sirex noctilio*, venom, wounding

## Abstract

**Simple Summary:**

*Sirex noctilio*, as a devastating international forestry quarantine pest whose venom can cause a series of physiological changes in the host plants, such as needle wilting, yellowing, decreased transpiration rate and increased respiration rate, etc. In this study, a full-length reference transcript of *Pinus sylvestris* var. *mongolica* was constructed by combining second- and third-generation transcriptome sequencing technologies. We also identified the specific expression genes and transcription factors of *P. sylvestris* var. *mongolica* under *S. noctilio* venom and wounding stress. *S. noctilio* venom mainly induced the expression of genes related to ROS, GAPDH and GPX, and mechanical damage mainly induced the photosynthesis−related genes. The results provide a better understanding of the molecular regulation of pine trees in response to *S. noctilio* venom.

**Abstract:**

*Sirex noctilio* is a major international quarantine pest that recently emerged in northeast China to specifically invade conifers. During female oviposition, venom is injected into the host together with its symbiotic fungus to alter the normal *Pinus* physiology and weaken or even kill the tree. In China, the Mongolian pine (*Pinus sylvestris* var. *mongolica*), an important wind-proof and sand-fixing species, is the unique host of *S. noctilio*. To explore the interplay between *S. noctilio* venom and Mongolian pine, we performed a transcriptome comparative analysis of a 10-year-old Mongolian pine after wounding and inoculation with *S. noctilio* venom. The analysis was performed at 12 h, 24 h and 72 h. PacBio ISO-seq was used and integrated with RNA-seq to construct an accurate full-length transcriptomic database. We obtained 52,963 high-precision unigenes, consisting of 48,654 (91.86%) unigenes that were BLASTed to known sequences in the public database and 4309 unigenes without any annotation information, which were presumed to be new genes. The number of differentially expressed genes (DEGs) increased with the treatment time, and the DEGs were most abundant at 72 h. A total of 706 inoculation-specific DEGs (475 upregulated and 231 downregulated) and 387 wounding-specific DEGs (183 upregulated and 204 downregulated) were identified compared with the control. Under venom stress, we identified 6 DEGs associated with reactive oxygen species (ROS) and 20 resistance genes in Mongolian pine. Overall, 52 transcription factors (TFs) were found under venom stress, 45 of which belonged to the AP2/ERF TF family and were upregulated. A total of 13 genes related to the photosystem, 3 genes related photo-regulation, and 9 TFs were identified under wounding stress. In conclusion, several novel putative genes were found in Mongolian pine by PacBio ISO seq. Meanwhile, we also identified various genes that were resistant to *S. noctilio* venom, such as GAPDH, GPX, CAT, FL2, CERK1, and HSP83A, etc.

## 1. Introduction

Mongolian pine (*Pinus sylvestris* var. *mongolica*), a geographical variety of Scots pine (*P. sylvestris*), is naturally distributed in the Daxinganling Mountains and the Honghuaerji of Hulunbuir in northeast China, as well as in some parts of Russia and Mongolia [1]. It exhibits a high tolerance to cold, drought, and low soil fertility, thereby exhibiting strong adaptability and rapid growth [2,3]. Owing to these characteristics, Mongolian pine is currently the main coniferous plant that is utilized for windbreaking and sand fixation in the 3-North area of China, and thus plays a role in environmental protection and ecological construction. However, *Sirex noctilio* Fabricius (Hymenoptera: Siricidae), an invasive pest, was first identified in Daqing, Heilongjiang Province, northeast China, which specifically invaded *P. sylvestris* var. *mongolica* and caused tremendous economic losses and ecological damage [4]. To date, it has been spread to several provinces in China [5,6].

*S. noctilio* is a wood-wasp species native to Eurasia and North Africa. In the native range, the females attack only weak or dying *Pinus* spp. and are thus considered to be a secondary pest of negligible economic or ecological impact [7,8,9,10]. By contrast, *S. noctilio* is a major pest in invasive sites that causes damage and even kills a large number of healthy pines; for example, it has destroyed 70% of *P. radiata* in Uruguay, 30% of *P. taeda* and *P. elliotii* in northeast Argentina, and 75% *P. ponderosa* and *P. contorta* in southwest Argentina [11]. Thus, *S. noctilio* has a variety of hosts, which can cause great economic losses to the forestry of the invasive sites.

The females inject venom and symbiotic fungus together with eggs as they oviposit into the xylem of the host pines [12,13,14,15]. The synergetic effect of the venom and fungus is lethal to the host, although the venom alone can weaken trees by causing needle wilt and yellowing [16,17]. Studies have shown that the venom induces a series of physiological changes in the host that weaken the host’s defense response and hence contribute to the growth of symbiotic fungus and the development of eggs [15]. These physiological changes include those related to needle dry weight, starch accumulation, peroxidase and amylase activity, respiration rate, and leaf pressure [18,19]. In addition, a few studies have used molecular methods to investigate the effects of the venom on the host plant. For example, in a study, the gene expression of pine tissue responding to *S. noctilio* venom was determined using a 26,496-feature loblolly pine cDNA microarray [20].

With the advancement of molecular-biology techniques, RNA-seq has become an indispensable tool for the analysis of defense gene expression and regulation in the transcriptome. The technique has been extensively used to study the response of conifers such as *P. halepensis* [21], *P. tecunumanii* [22], and *Picea sitchensis* [23] to biotic and abiotic stresses. However, for species lacking a reference genome, most genes obtained by RNA-seq were assembled from short reads. In this study, we obtained the full-length transcriptome of *P. sylvestris* var. *mongolica* for the first time by using the PacBio sequencing approach. Furthermore, we systematically identified and analyzed the functions and signaling pathways of differentially expressed genes (DEGs), and the molecular mechanisms of needles in response to venom and wounding stress. The study results may deepen our understanding of the molecular mechanism of the *P. sylvestris* var. *mongolica* resistance to *S. noctilio* venom, as well as help in identifying resistance-related genes and promoting genetic improvement.

## 2. Materials and Methods

### 2.1. Venom and Plant-Material Collection and Inoculation Trial

*S. noctilio* females obtained from the Biosafety Laboratory in Beijing Forestry University (BFU, Beijing, China) were immediately stored at −80 °C after their emergence from pine (*P. sylvestris* var. *mongolica*) logs infested by *S. noctilio*. Frozen wood wasps were dissected on ice to isolate the venom sac under a microscope (×40) (Leica M205C, Heidelberg, Germany). The pooled venom sac was added into a 1.5 mL centrifuge tube with four steel balls (diameter = 0.4 mm), and the mixture was shaken three times (2 min each time, frequency = 40/s). Then, the mixture was diluted to a 20 mg/mL concentration in a solution with deionized water. The resultant solution was centrifuged at 16,000× *g* and 4 °C for 30 min to remove cell debris, and the supernatant was stored at −20 °C until inoculation.

The inoculation experiments were conducted in approximately 10-year-old planted *P. sylvestris* var. *mongolica*, which were derived from a single seed lot and located in Ergetu Pine Plantation of Ulanhot CT, northeast China. Nine healthy pine trees with similar heights (174 ± 9 cm) and diameters at breast height (3.82 ± 0.22 cm) were used in the experiment that involved three treatments, namely control (non-inoculated), inoculated, and wounded. For inoculated and wounded samples, two 1.5 cm-deep holes were drilled at the 20 cm and 25 cm heights of the trunk with a cordless drill (6 mm diameter) and angled down by 45 degrees in order to hold a total of 2 mL of 20 mg/mL *S. noctilio* venom or water. Each hole was wrapped with parafilm strips. 3–5 needles were cut with sterile scissors in 4 directions (east, south, north and west) on the annual shoots at the top of each pine tree. The leaf needles were collected at three time points (0 h, 24 h, and 72 h) and immediately placed in liquid nitrogen and brought back to the laboratory for storage at −80 °C until RNA extraction.

### 2.2. PacBio Iso-Seq Library Preparation and Sequencing

A total of 27 samples (3 treatments: control, inoculation, wounding × 3 time points: 0 h, 24 h, 72 h × 3 biological replicates) were used for RNA sequencing. Total RNA was extracted from needles by using the E.Z.N.A. Plant RNA Kit (OMEGA bio-tek, Norcross, GA, USA), according to the manufacturer’s protocol, and the RNA quality and concentration were measured using the NanoDrop 2000 Spectrophotometer (NanoDrop Products, Rockville, MD, USA). The RNA integrity number was assessed using Agilent 2100 Bioanalyzer (Agilent Technologies, Wilmington, DE, USA). The quality assessment of the 27 samples used for sequencing is shown in Appendix A. Equal amounts of RNA from 27 samples (1 μg per sample) were pooled together to form total RNA, and then the SMART library was prepared using 5 μg total RNA. After PCR amplification, the products were used to construct the SMRTbell template library using the Iso-Seq protocol. Then, the libraries were prepared for sequencing by annealing a sequencing primer and binding polymerase to the primer-annealed template.

### 2.3. Iso-Seq Data Analysis

The raw polymerase reads were processed using the SMRT Pipe analysis workflow with default parameters. Next, CCSs were obtained through conditional screening (full passes of 1 and quality of 0.9) from the subread files. The CCSs were further classified into FL and non-full-length (nFL) transcripts irrespective of the presence of the 5′ primer sequence, 3′ primer sequence, and a poly-A tail. Full-length non-chimera (FLnc) reads were subjected to isoform-level clustering, followed by arrow polishing with the nFL sequence (hq_quiver_min_accuracy 0.99, bin_by_primer false, bin_size_kb 1, qv_trim_5p 100, qv_trim_3p 30). These polished sequences were further corrected, and redundant sequences were removed using the CD-HIT software (-c 0.99 -G 0 -aL 0.00 -aS 0.99 -AS 30 -M 0 -d 0 -p 1) to obtain the non-redundant, non-chimeric, and full-length transcripts for subsequent analysis (https://github.com/weizhongli/cdhit/wiki/3, accessed on 12 March 2022).

### 2.4. Illumina Library Preparation and Sequencing

The Illumina library was constructed using the TruseqTM RNA sample prep Kit (Illumina, San Diego, CA, USA). Briefly, polyadenylated mRNA was randomly disrupted into fragments. Under the action of reverse transcriptase, first-strand cDNA was generated using random hexamer primers (Invitrogen, Carlsbad, CA, USA), followed by second-strand cDNA synthesis. The purified fragmented cDNA was subjected to end-repair, followed by A-tailing. The final cDNA library was obtained by PCR enrichment and quantified using the TBS-380 (Tuner Biosystems, Sunnyvale, CA, USA). Finally, Illumina NovaSeq 6000 was used for sequencing.

### 2.5. Gene Functional Annotation

Full-length transcripts were annotated by performing BLASTX searches against six public databases, namely NR, KEGG, COG, Pfam, and Swiss-Prot, using the Blast2GO program for GO annotation based on NR annotation. The cut off E-value < 10^−5^ was used in BLAST analysis against these databases.

### 2.6. Quantification of Unigene Expression Levels and Analysis of DEGs

To determine the difference in the gene expression of *P. sylvestris* var. *mongolica* under different treatments, we identified the gene-expression level of each sample by using RSEM software with the default parameters [24]. All clean data generated by Illumina sequencing were mapped to the full-length transcripts, and the read count of each gene was obtained in *P. sylvestris* var. *mongolica*. All the read-count values were converted into the TPM value to calculate the expression of each gene [25].

DEG analysis between the venom-stressed and control samples and between the wounding-stressed and control samples was performed using the DESeq R package (1.10.1) [26]. Significant DEGs were assigned with thresholds based on the false-discovery rate (FDR) < 0.05 and log2|fold change| ≥ 1. GO and KEGG enrichment analyses for all DEGs were performed using the Python goatools package and KOBAS software. The potential TFs of DEGs were identified using the BLAST method with the PlantTFDB 4.0 database.

## 3. Results

### 3.1. Full-Length Transcriptome Sequencing

The RNA of all samples was mixed for third-generation sequencing (TGS) by using the PacBio Sequel platform. A total of 6,588,055 subreads were generated by filtering the raw data with an average length of 3625 bp and an N50 value of 4581 bp. Then, we obtained 849,415,352 bp circular consensus sequences (CCS) (199,841 reads of insert) with an average length of 4250 bp through conditional screening (full passes ≥ 1, quality > 0.90). The number of full-length (FL) and full-length nonchimeric (FLnc) reads was 163,130 and 162,036, respectively, with the corresponding average length of the two reads being 4232 and 4184 (Table 1).

The advantage of the TGS technology by using the PacBio platform is that it provides long read lengths, although its single-base-error rate is high. To further improve the accuracy, we also sequenced 27 samples of the needles by using the Illumina NovaSeq 6000 platform (Illumina, Sam Diego, CA, USA) with 300 bp pair-end reads. Redundant and similar sequences were removed using CD-HIT software. Finally, 52,963 unigenes with an average length of 1801 bp were obtained and considered the reference transcriptome, and the length distribution of unigenes is shown in Appendix A.

### 3.2. Functional Annotation of the Full-Length Reference Transcriptome

To determine the possible functions of unigenes in *Pinus*, we analyzed a total of 52,963 unigenes by using six databases, namely NCBI nonredundant protein (NR), Gene Ontology (GO), Kyoto Encyclopedia of Genes and Genomes (KEGG), Clusters of Orthologous Groups of proteins (COG), Swiss-Prot, and Protein Family (Pfam). Overall, 48,654 (91.86%) unigenes were annotated to known sequences in public database, whereas 20,172 (38.09%) unigenes were simultaneously annotated in six databases. As shown in Figure 1A, the number of unigene hits was the highest (48,228; 91.06% and 44,087; 83.24%) in the Nr database, followed by those in the GO database (41,580; 78.51%), Swiss-Prot database (39,332; 74.26%), Pfam database (37,874; 71.51%), and KEGG database (25,157; 47.50%). According to the sequence alignment in the NR database, 24,848 (51.52%) sequences had significant homology against *P. sitchensis*, followed by those against *Amborella trichopoda* (3462, 7.18%), *Nelumbo nucifera* (1489, 3.09%), *Cinnamomum micranthum* (1364, 2.83%), *Macleaya cordata* (775, 1.61%), *P. taeda* (736, 1.53%), *Vitis vinifera* (611, 1.27%), *P. tabuliformis* (608, 1.26%), *P. sylvestris* (600, 1.24%), and *Elaeis guineensis* (570, 1.18%). Nevertheless, 28.47% of the sequences were homologous to those of other species (Figure 1B).

For the GO analysis, 41,580 unigenes were divided into 52 categories, namely molecular function (MF, 33,774), cellular component (CC, 27,629), and biological process (BP, 22,787). Of the total, 16 categories belonged to MF, 14 categories belonged to CC, and 22 categories belonged to BP (Figure 2A). To further profile the pathways in which the unigenes are involved, we conducted an analysis based on the KEGG database. A total of 25,157 unigenes were classified into five metabolic pathways (first category) and were found to be mainly associated with carbohydrate metabolism (3057, 12.15%), followed by energy metabolism (2445, 9.72%) and translation (2151, 8.55%) (Figure 2B).

Furthermore, we also predicted 1017 unigenes, which were assigned to the 31 TF family. The three most abundant unigenes were MYB (170), AP2/ERF (130) and NAC (87) (Appendix A).

### 3.3. Identification of DEGs

Libraries obtained from the control, wounded, and inoculated samples were mapped to the reference transcripts from the PacBio ISO-seq. The matched rate of all the clean reads was >80% (Appendix A). The difference in the expression levels between two groups was determined based on the transcripts-per-million (TPM) value. Samples that were inoculated with venom, wounded, and those without any treatment were replaced with PI, PW and CK, respectively. For inoculation, we identified a total of 266 DEGs (139 upregulated and 127 downregulated) between PI_0 hand PI_24 h, 995 DEGs (802 upregulated and 193 downregulated) between PI_24 h and PI_72 h, and 1253 DEGs (905 upregulated and 348 downregulated) between PW_0 hand PI_72 h. For wounding treatment, we identified 301 DEGs (111 upregulated and 190 downregulated) between PW_0 h and PW_24 h, 365 DEGs (258 upregulated and 107 downregulated) between PW_24 h and PW_72 h, and 1,193 DEGs (602 upregulated and 591 downregulated) between PW_0 h and PW_72 h. These results show that the number of DEGs increased with an increase in the treatment time. For example, the number of DEGs after 72 h of inoculation was 1253, which was approximately five times higher than that after 24 h of inoculation (266) (Figure 3).

### 3.4. Analysis of DEGs in Wounded and Inoculated Mongolian Pine at 72 h

Among the three time points, the specific-treatment time point was 72 h, at which the DEGs were most abundant. Therefore, the 72 h time point was used for subsequent treatments. Compared the CK_72 h, 825 DEGs were induced by *S. noctilio* venom, 506 DEGs were induced by wounding, and 119 DEGs were co-induced by venom and wounding.

#### 3.4.1. Analysis of Inoculation-Specific DEGs

Among the 825 DEGs between control and inoculation, 559 genes were upregulated, and 266 genes were downregulated. A total of 706 inoculation-specific DEGs (474 upregulated and 232 downregulated) were identified after removing the DEGs common to wounding and inoculation (Appendix A).

To investigate the molecular mechanisms that regulate interactions and coordination of pine under *S. noctilio* venom-specific presentation, a GO enrichment analysis (top 20) of these 706 DEGs was performed, which indicated that inoculation affected 12 MF categories, as well as seven BP and one CC. The most significantly enriched MF categories were “ADP binding” (31 upregulated and 3 downregulated), “transcription-regulator activity” (46 upregulated and 1 downregulated), and “DNA-binding transcription-factor activity” (45 upregulated and 1 downregulated). Only one CC category was found to be significantly enriched, namely “nucleus” (68 upregulated and 12 downregulated), whereas “signal transduction” (45 upregulated and 1 downregulated), “xyloglucan metabolic process”, and “hemicellulose metabolic process” were the most significantly enriched BP categories. Notably, all six genes enriched in the “xyloglucan metabolic process” and “hemicellulose metabolic process” categories were upregulated and identical (Figure 4A). Of these, five genes were annotated as “xyloglucan endotransglucosylase/hydrolase 2”, whereas one gene was annotated as “unknown” in the NR database.

Furthermore, the KEGG-pathway enrichment analyses of 706 DEGs were performed, and these DEGs were assigned to 80 KEGG pathways. The top 20 pathways were screened as having the most intensive response activities. The most significantly enriched pathways were “plant–pathogen interaction”, “arginine biosynthesis”, “biosynthesis of unsaturated fatty acids”, “necroptosis”, and “fatty-acid elongation” (Figure 4B).

#### 3.4.2. Analysis of Wounding-Specific DEGs

A total of 506 DEGs were identified by comparing the gene-expression levels under wound treatment with those under control treatment. A total of 387 DEGs were differentially expressed specifically after wounding; of these, 183 DEGs were upregulated and 204 DEGs were downregulated. GO enrichment analysis (top 20) of the 387 DEGs showed that wounding affected 4 MF, 13 CC, and 3 BP categories. Seven exactly identical genes were assigned to the most significantly enriched MF categories, namely “inositol 3-alpha-galactosyltransferase activity”, “glucuronosyltransferase activity”, and “UDP-galactosyltransferase activity”. Six DEGs were assigned to the most significantly enriched CC categories, namely “photosystem II antenna complex”, “PSII-associated light-harvesting complex II”, and “light-harvesting complex”. The significantly enriched BP categories were “photosynthesis”, light harvesting”, “photosystem II assembly”, and “nonphotochemical quenching” (Figure 5A).

The wound-specific DEGs were also subjected to KEGG enrichment analysis and were assigned to 83 metabolism pathways. The top 20 significantly enriched pathways are shown in Figure 5B and were divided into four categories, namely “metabolism (M)”, “cellular processes (CP)”, “genetic information processing (GIP)”, and “organismal systems (OS)”. The most significantly enriched M categories were “photosynthesis-antenna proteins”, “galactose metabolism”, “pentose and glucuronate interconversions”. In CP and GIP, only one category was enriched, namely “RNA degradation” and “necroptosis”, respectively.

#### 3.4.3. DEGs Induced by Inoculation and Wounding

A total of 119 common DEGs (85 upregulated and 34 downregulated) were induced by inoculation and wounding compared with the control. The top 20 categories of significant GO enrichment are shown in Figure 6A. Interestingly, seven DEGs were simultaneously enriched in “amide binding” and “peptide binding”. Four of these seven DEGs were enriched in “chloroplast-targeting sequence binding”, “intrinsic component of chloroplast outer membrane”, and “protein targeting to chloroplast”. However, the annotation information of these seven genes is “unknown” in the NR database. Subsequently, the KEGG enrichment analysis showed that the most significantly enriched categories were “nitrogen metabolism”, “photosynthesis-antenna proteins”, and “glutathione metabolism” (Figure 6B). Notably, most of the categories were related to the chloroplast and photosynthesis.

### 3.5. Transcription Factors (TFs) in Response to Inoculation and Wounding in P. sylvestris var. mongolica

Of the 706 DEGs specifically induced by *S. noctilio* venom, we identified 7 TF families with 52 DEGs (50 upregulated and 2 downregulated). Among these TFs, the largest number of genes (46) belonged to the ethylene-responsive-factor (AP2) family, with 45 genes being upregulated and 1 gene being downregulated. The number of other TFs, including Zf-CCCH, Zf-C2H2, Myb_DNA_bind, GRAS, DUF573, and DUF260, was small. For wounding-specific DEGs, four TFs were identified with only nine DEGs. The number of genes for AP2, Myb_DNA_bind, and NAM and DUF260 was 4, 3, and 1, respectively. Of the 119 DEGs induced by inoculation and wounding, only 6 DEGs belonged to the three TFs, AP2, Myb_DNA_bind, and DUF260 (Table 2).

## 4. Discussion

In the past decade, short-read RNA-seq (next-generation sequencing) has been widely used for genetic discovery and research [27]. However, a limitation of the second-generation technologies is that the length of sequenced fragments is usually less than 300 bp [28]. Moreover, obtaining a complete and reliable reference for species without reference genomes is challenging. The third-generation sequencing technique, that is, single-molecule sequencing, which can be used to generate full-length transcripts and near-reference-genome assembly for some species, can reveal the full structure of individual transcripts [29,30]. The present study is the first to conduct transcriptome sequencing of *P. sylvestris* var. *mongolica* needles by using the single-molecule-sequencing technique. A total of 52,963 full-length unigenes (average length, 3625 bp; N50 value, 4581 bp) were generated, which is more than that generated in a previous study (39,231) (N50 value of 1646 bp) [31]. Additionally, the annotated percentage (48,654, 91.86%) of these full-length unigenes was significantly higher than that reported in previous studies [32,33]. The remaining 4309 (8.14%) unannotated unigenes possibly represent a novel gene pool that is specific to the common *P. sylvestris* var. *mongolica*. The potential function of these unannotated genes and their response to *S. noctilio* venom or wounding warrant further investigation.

### 4.1. Signaling Pathways Involved in the Response of P. sylvestris var. mongolica to Wounding and Inoculation

Plant responses to biological stress are extremely complex because of the presence of various types of interactions between the plant and the pathogen [34]. These interactions serve as the multi-layered immune system that has emerged during plant evolution to prevent or block the colonization of most potential pathogens [35,36]. This in turn promotes the production of specific resistant proteins in plants to recognize pathogen/insect effects [37]. Consistently, in the present study, we found the “plant–pathogen interaction” as the most significant KEGG pathway during the 72 h inoculation (Appendix A). Among these genes, those encoding CAM/CML were the most significantly upregulated. During the plant–pathogen interaction, calcium is considered to be an important factor for regulating plant responses to various pathogens and herbivores [38]. Ca2+ signaling by CaM/CML produces NO, which induces defense responses [39]. The CaM/CML protein family regulates cellular responses to various stimuli in grapevine, particularly to biotic stresses [40]. Trujillo-Moya et al. reported a similar conclusion on *P. abies* [41]. In addition, the genes encoding FLAGELLIN-SENSING 2 (FL2), CERK1, and HSP83A were upregulated, all of which have been found to play crucial roles in biotic and abiotic stresses (Table 3) [42,43,44].

Photosynthesis is the most fundamental and complex physiological process in all green plants, and this process is also severely affected by various biotic and abiotic stresses [45]. The photosynthetic pigments are believed to be damaged by stress factors, which in turn reduces the photosynthetic capacity by reducing the light-absorption efficiency of the photosystem (PSI and PSII) [46,47]. The KEGG-pathway enrichment analysis of the DEGs induced by mechanical specificity and the co-induced DEGs showed that the most significantly enriched pathways were “photosynthesis−antenna proteins” and “nitrogen metabolism”. The genes involved in the nitrogen-metabolism pathway were annotated as carbonic anhydrase and glutamine synthetase cytosolic isozyme, both of which were upregulated (Appendix A). Carbonic anhydrase is an essential enzyme in photosynthesis that exhibits the property of reversibly converting CO_2_ to HCO_3_^−^ [48]. In general, wounding mainly induces the differential expression of genes related to photosynthetic metabolic pathways in *P. sylvestris* var. *mongolica*.

### 4.2. Expression of Typical Venom-Induced Genes in Mongolian Pine

Under stress, plants regulate their homeostatic mechanisms by generating excess ROS, and have evolved with some enzymatic or non-enzymatic detoxification mechanisms to control the excess accumulation of ROS [49,50,51]. In our study, a total of 706 genes were differentially expressed specifically under venom stress. Of these, we found six DEGs associated with reactive oxygen species (ROS), of which three encode glyceraldehyde-phosphate dehydrogenase (GAPDH), two encode glutathione peroxidase (GPX) and one encodes catalase (CAT). In addition, we also found five DEGs encoding Xyloglucan endotransglucosylase/hydrolase 2 (XTH2), five putative truncated TIR-NBS-LRR protein, four polyubiquitin, three UDP-glycosyltransferase (UGT), two Hs1pro-1 and one pleiotropic drug-resistance (PDR) protein 1. One of these was a downregulated DEG, and the others were upregulated DEGs. These genes have been shown to be involved in regulating plant immune responses to biological or abiotic stresses, e.g., XTH expression could be rapidly induced in response to various stresses in tobacco [52]. The C-termini of the five putative truncated TIR-NBS-LRR proteins contain a variable LRR domain, which is the most significant structural feature of plant disease-resistance proteins [53]. All of these genes exhibited significantly induced expression 72 h after venom stress, but the expression of these genes was not significant under wounding stress (Table 4). As a related species of *S. noctilio*, *Sirex nitobei* venom proteins have been identified, such as laccase-2, laccase-3, a protein belonging to the Kazal family, chito-oligosaccharidolytic β-N-acetylglucosaminidase, beta-galactosidase, etc. These venom proteins may be the key factors to inducing gene-specific expression in *P. sylvestris* var. *mongolica* [54]. Therefore, the DEGs induced by *S. noctilio* venom were identified as the candidate genes that enhance the self-tolerance of *P. sylvestris* var. *mongolica*, especially for *S. noctilio*. We recommend that researchers study these genes in more detail.

### 4.3. Expression of Typical Wounding-Induced Genes in Mongolian Pine

Under wounding stress, the results showed that 387 DEGs were specifically regulated. Photosynthesis is essential for plant growth and development. It involves the harvesting of light and transfer of solar energy, using light-harvesting chlorophyll-a/b-binding (LHC) proteins. The LHC proteins are among the most abundant in thylakoids, which are encoded by nuclear genes. In high plants, LHC proteins include a large gene family containing 10–12 members, constituting the peripheral light-harvesting antenna of photosystem I (PSI) and photosystem II (PSII). LHCAs are encoded by Lhca1–Lhca6, and LHCBs are encoded by Lhcb1–6 [55,56,57,58]. We found 13 DEGs associated with photosynthesis, 6 encoding Lhcb5 protein, 4 encoding LHC protein, partial, 2 encoding oxygen-evolving enhancer protein 1 (OEE) and 1 encoding Lhca4 protein, 5 of which were upregulated and the others were downregulated (Table 5). Many studies have indicated that LHC gene expression is regulated by multiple environmental and developmental cues in higher plants, such as light, oxidative stress, low temperatures, and phytohormone abscisic acid [59,60,61,62]. In addition, we also found three casein kinase 1-like protein HD16s, which can change the alteration of photoperiod sensitivity to enhance the adaptability to local environmental conditions in many plants [63]. Therefore, we hypothesize that these genes still function after the plants are subjected to wounding stress.

### 4.4. The Role of TFs in Response to Inoculation and Wounding

Families of TFs, as activators and repressors, play an important role in plant resistance to biotic and abiotic stresses [64,65]. In higher plants, approximately 60 TF families have been identified [66], which include AP2/ERF, MYB, and C2H2. The APETALA2/ethylene-responsive element-binding factors (AP2/ERF) represent a large group of factors that are found mainly in plants [67,68,69]. AP2/ERF are the important regulators involved in plant growth and development [70,71], hormonal regulation, and abiotic stress [72]. They also play a crucial role in plants’ defense against biotic stress, including invasion by herbivorous insects and microbial pathogens [73]. In this study, we identified 46 (88.46%) DEGs (45 upregulated and 1 downregulated) associated with the AP2/ERF family, all of which were annotated as ERF in the Swiss-Prot database. However, only four AP2/ERFs were present in wound-specific and co-induced DEGs. The ERF TFs enhance plant resistance to piercing/sucking or chewing insects by stimulating the accumulation of SA, JA, and H2O2 [74]. Thus, *P. sylvestris* var. *mongolica* mainly upregulated the genes encoding ERF in response to *S. noctilio* venom stress. In addition, three upregulated TFs and one downregulated TF, belonging to the GRAS, C3H, C2H2, and GeBP families, respectively, were induced specifically under venom stress. Previous studies have reported that the TF *A. thaliana* GeBP-LIKE 4 was rapidly induced in root tips in response to toxic metals [75]. GRAS-family proteins are present only in plants and not in any other organisms, suggesting that these proteins represent an important and diverse set of regulatory molecules [76]. Moreover, the C3H-family proteins have been reported to participate in lignin synthesis in plants, and the C3H gene downregulation could reduce the lignin content in rice straw. Taken together, most of the TF family members exhibited an inducible expression profile under venom stress in *P. sylvestris* var. *mongolica*, which indicates that these TFs play a vital role in modulating needle signal transduction and molecular regulation.

## 5. Conclusions

In this study, 4309 (8.14%) new genes of *P. sylvestris* var. *mongolica* were obtained by combining the PacBio ISO-seq and Illumina RNA-seq technology. Under *S. noctilio* and wounding stress, 825 and 506 genes were differentially expressed, respectively, and 119 DEGs were present in both types of stresses. This information provides additional resources to identify and unearth important genes in *P. sylvestris* var. *mongolica*. Inoculation-specific and wounding-specific DEGs were significantly enriched in “plant–pathogen interaction” and “photosynthesis-antenna proteins”. In addition, *S. noctilio* venom mainly caused the differential expression of pine resistance genes, such as GAPDH, GPX, CAT, FL2, etc. However, wounding only caused the differential expression of photosynthesis-related genes such as Lhcb5 protein, LHC protein and OEE, etc.

## Figures and Tables

**Figure 1 insects-13-00338-f001:**
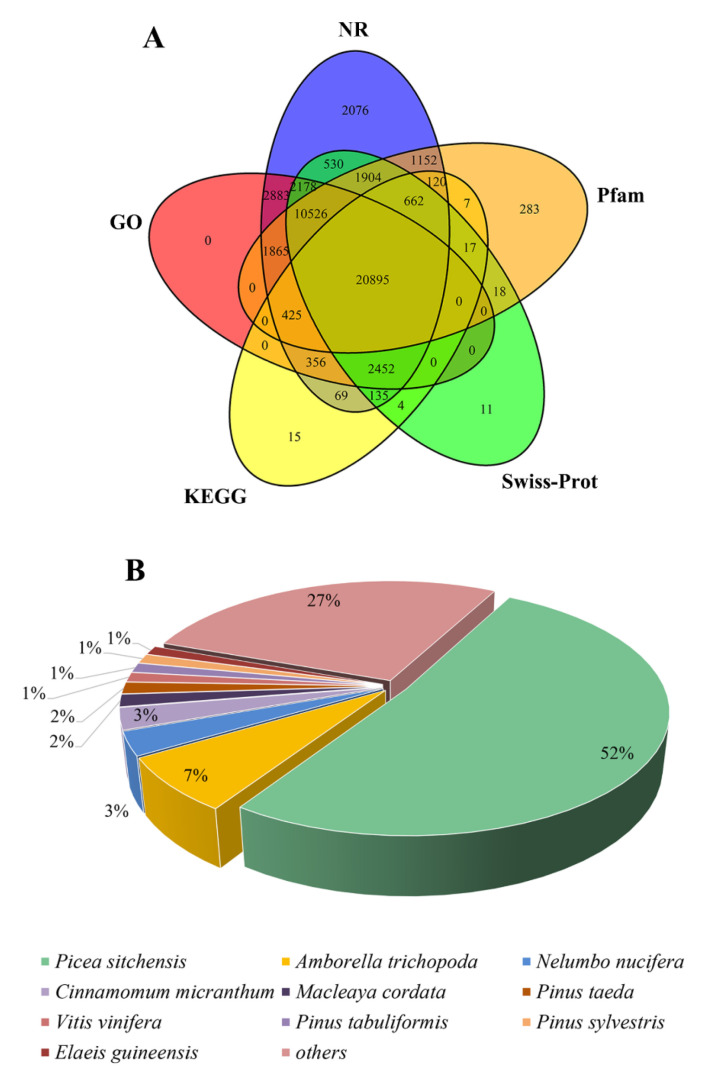
Unigene annotation of PacBio sequencing. (**A**) Overlap between the number of all unigenes according to five databases. (**B**) Distribution of unigene annotations based on the NR database for the species-distribution statistics.

**Figure 2 insects-13-00338-f002:**
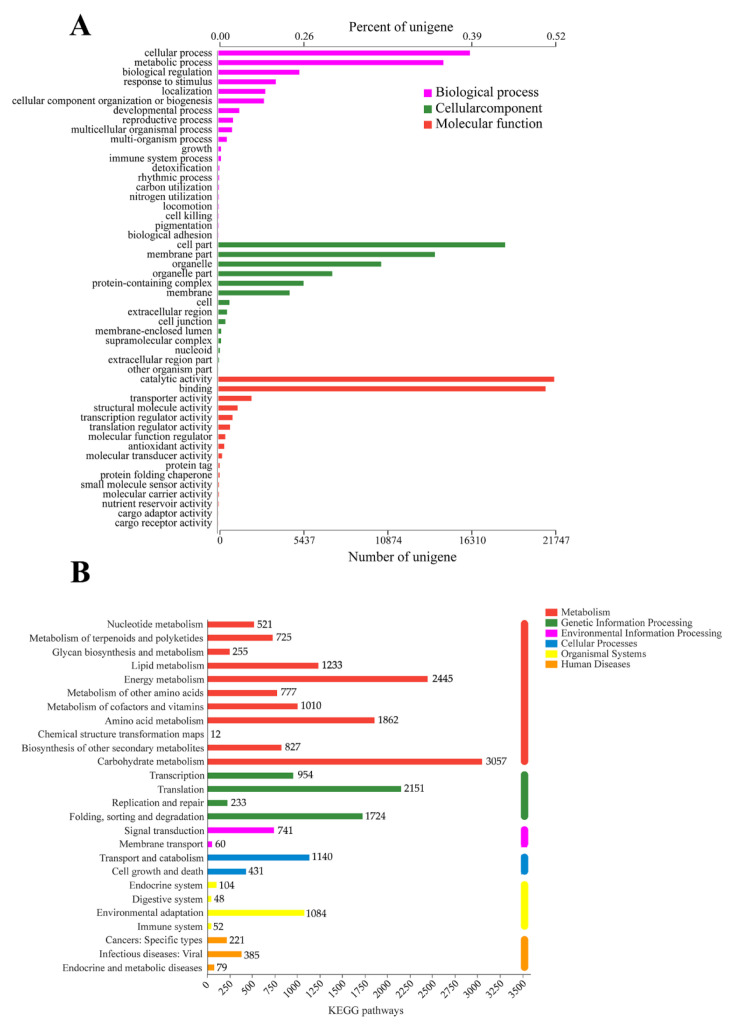
GO and KEGG classification of unigenes. (**A**) GO functional classification of all unigenes. (**B**) KEGG classification of all unigenes.

**Figure 3 insects-13-00338-f003:**
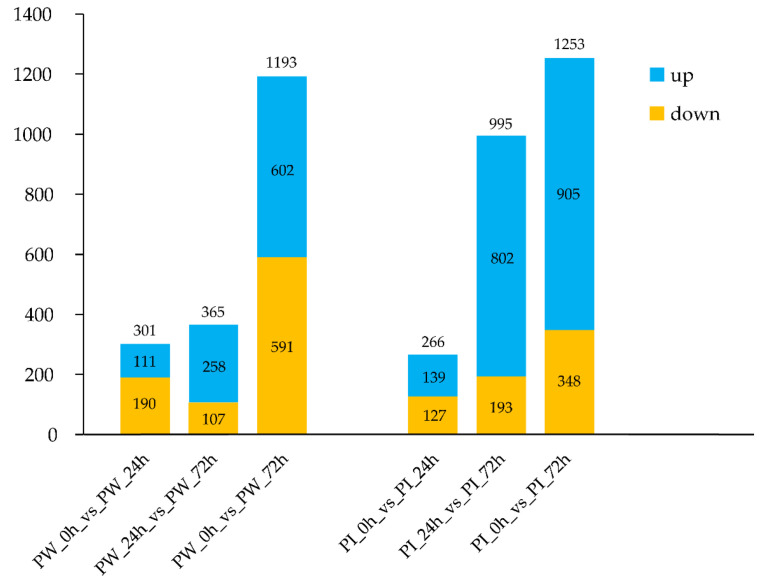
Number of DEGs induced by wounding and inoculation at different times.

**Figure 4 insects-13-00338-f004:**
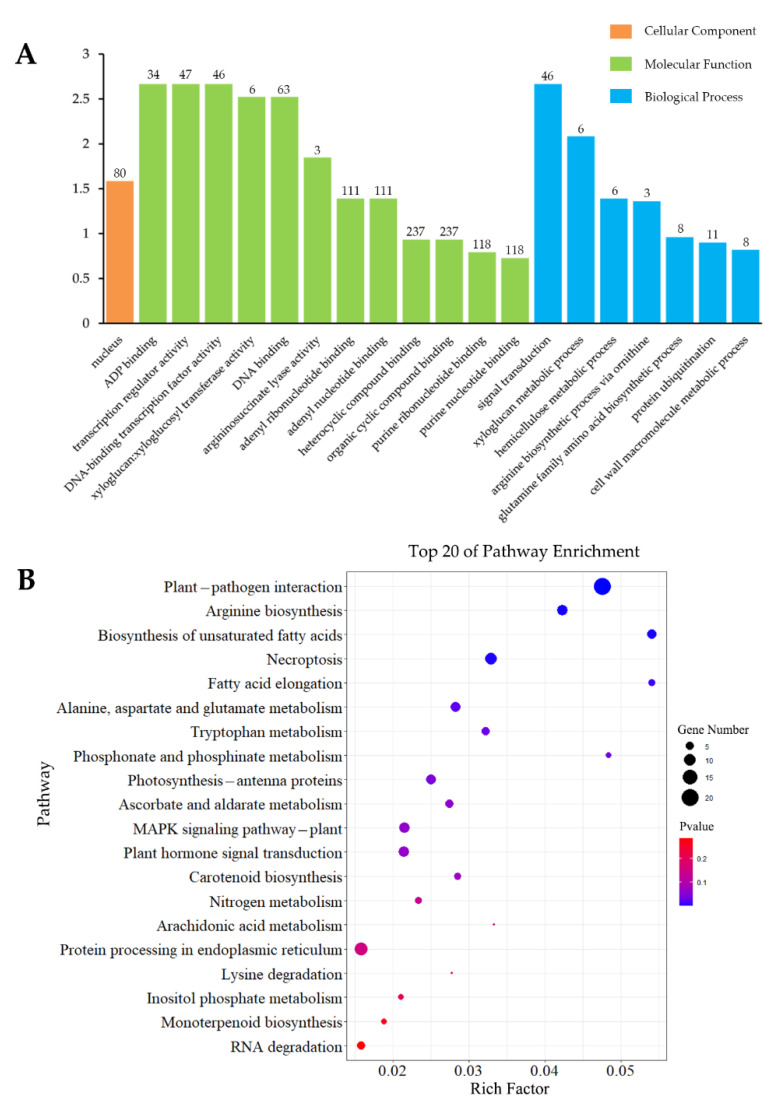
GO (**A**) and KEGG (**B**) analyses of DEGs induced by inoculation specificity.

**Figure 5 insects-13-00338-f005:**
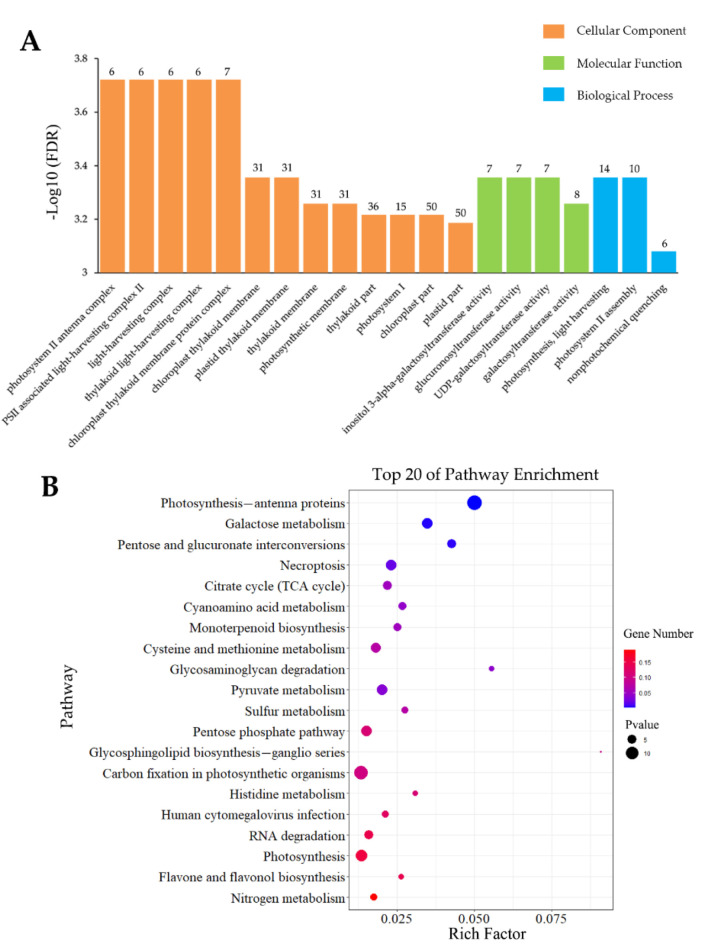
GO (**A**) and KEGG (**B**) analysis of DEGs induced by wounding specificity.

**Figure 6 insects-13-00338-f006:**
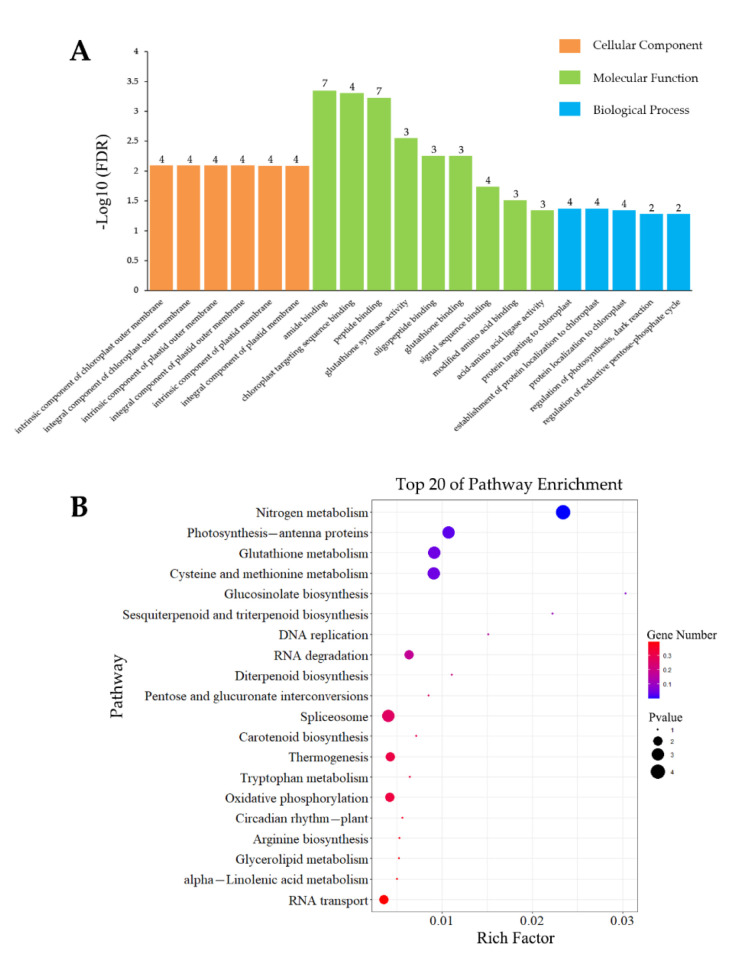
GO (**A**) and KEGG (**B**) analysis of DEGs co-induced by inoculation and wounding.

**Table 1 insects-13-00338-t001:** Summary of the transcriptome data from the PacBio platform.

Type	Number
Subread Reads	6,588,055
Average Subread Length	3625
Subread N50	4581
Total CCS	199,841
Average CCS Length (bp)	4250
FL Reads	163,130
FL Mean Length (bp)	4232
FLNC Reads	162,036
FLNC Mean Length (bp)	4184
Total Unigenes	52,963
Average Unigene Length (bp)	1801
TF number	1017

**Table 2 insects-13-00338-t002:** TF statistics of DEGs under different treatment.

TF Family	Number of TFs		
	Inoculation-Specific	Wounding-Specific	Co-Induced
AP2/ERF	46	4	4
LOB	1	1	1
MYB_superfamily	1	3	1
C2H2	1	N/A	N/A
C3H	1	N/A	N/A
GeBP	1	N/A	N/A
GRAS	1	N/A	N/A
NAC	N/A	1	N/A

**Table 3 insects-13-00338-t003:** The key DEGs involved in plant–pathogen interaction signaling pathway under the *S. noctilio* venom stress.

Gene_ID	Swiss-Prot	Log2FC(PI_72 h/CK_72 h)	*p*-Value
transcript_26258	Probable disease-resistance protein At1g15890 *Arabidopsis thaliana*	1.84	2.31 × 10^−6^
transcript_64802	Calmodulin-like protein 3 *Arabidopsis thaliana*	4.40	3.95 × 10^−7^
transcript_66692	Putative calmodulin-3 (Fragment) *Solanum tuberosum*	4.78	2.69 × 10^−8^
transcript_11085	Calmodulin-like protein 2 *Arabidopsis thaliana*	4.36	1.26 × 10^−15^
transcript_52365	LRR receptor-like serine/threonine-protein kinase FLS2 *Arabidopsis thaliana*	4.24	5.83 × 10^−5^
transcript_35906	Heat-shock protein 83 *Ipomoea nil*	2.25	2.47 × 10^−5^
transcript_25250	Heat-shock protein 83 *Ipomoea nil*	4.02	3.52 × 10^-4^

**Table 4 insects-13-00338-t004:** DEGs expression under the *S. noctilio* venom stress of Mongolian pine.

Annotation	Gene_ID	Log2FC(PI_72h/CK_72h)
GAPDH	transcript_53182	7.7805
	transcript_44911	4.5701
	transcript_1666	3.1995
GPX	transcript_43923	2.1892
	transcript_54716	1.5788
CAT	transcript_40212	3.5800
XTH2	transcript_48021	4.9165
	transcript_59698	3.7113
	transcript_7449	3.3294
	transcript_52674	2.7133
	transcript_7326	2.5641
UGT	transcript_70636	2.6561
	transcript_40291	1.7841
	transcript_41406	1.6345
putative truncated TIR-NBS-LRR protein	transcript_20771	2.7826
	transcript_19948	2.1045
	transcript_19947	1.9758
	transcript_21018	1.7106
	transcript_20174	1.5549
polyubiquitin	transcript_65149	9.5582
	transcript_55582	2.1239
	transcript_5903	−1.554
	transcript_6253	7.3522
PDR protein 1-like	transcript_37577	6.2179
Hs1pro-1	transcript_23455	2.1044
	transcript_65743	2.1014

**Table 5 insects-13-00338-t005:** DEGs expression under wounding stress of Mongolian pine.

Annotation	Gene_ID	Log2FC(PW_72 h/CK_72 h)
Lhcb5 protein	transcript_58449	10.3402
	transcript_61176	9.1811
	transcript_71238	5.2818
	transcript_8414	2.5689
	transcript_67898	−7.5324
	transcript_15721	−9.1712
LHC protein, partial	transcript_73156	7.4302
	transcript_67949	−2.3693
	transcript_51999	−2.3040
	transcript_47491	−2.4656
OEE	transcript_7777	−4.6929
	transcript_5485	−2.0221
Lhca4 protein	transcript_49364	−2.3063
HD16	transcript_28611	−7.1177
	transcript_31667	−9.0054
	transcript_17461	−9.0169

## Data Availability

The raw sequences have been deposited at SRA-NCBI (Accession Number: PRJNA 795839).

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
