# Peer review of "Full-Length Transcriptome Sequencing-Based Analysis of Pinus sylvestris var. mongolica in Response to Sirex noctilio Venom"

_insects, 2022, doi:10.3390/insects13040338_

Round 1

Reviewer 1 Report

In this manuscript, the authors used PacBio ISO-seq and Illumina sequencing technologies to identify and compare transcript expressions in Pinus sylvestris var. mongolica challenged with Sirex noctilio venom. Although the results presented is novel and justify a worthy scientific contribution, the manuscript is not well presented. Some sentences are not properly constructed and can lead to confusion or misunderstanding. The material and method section was not clearly presented. Some details on the analyses were omitted, no references to the programmes used were provided. There are also formatting errors throughout that should be fixed. I have attached the PDF copy of the manuscript with annotated comments and gave some general comments below to help the authors improve the manuscript.

L22: were blasted to at least one public database => were BLASTed to known sequences in public databases.

L28-L29: related to…

Section 2.1: Provide more information on how the needles were collected: How many needles per replicate and how were they collected: how far from the treatment point, at what stages (how old)? and from which side of the tree?

Section 2.3:

L 120-L121: The CCSs were further classified into FL and non-full-length (nFL) transcripts irrespective of the presence of the 5’ primer sequence, 3’ primer sequence, and a poly-A tail => If the RNA was degraded (clearly some would be), full length CCS reads do not equal to full-length transcripts. To obtain the full-length transcripts, it is important that RNA samples used for ISO-Seq sequencing are of very high quality – thus please provide the RIN values of the samples that were used for ISO-Seq and Illumina sequencing.

References to software used for isoform-level clustering, arrow, and CD-HIT etc need to be provided.

Section 2.5: Please provide more information on the parameters used. For example, what were the E-value, sequence coverage cut-offs used in BLAST analyses against these databases?

Section 2.6: Was illumina sequencing done for all 27 samples separately? How were read mapping and counting done? Provide reference to relevant tools and databases used.

Reviewer 2 Report

The subject of the manuscript is very interest. Authors provide a large amount of new data on the transcriptomic profiles of Mongolian pine responding to Sirex noctilio venom. The manuscript is well written in general though the language and grammar can be polished further to improve the manuscript.

However, the following comments and/or questions need to be addressed while revising it:

  1. Are three time points (12, 24, and 36 hours) the best time for collecting samples, reflecting the reprogramming of gene expression in the host plants are vigorously happening at that time? Authors has indicated that DEGs appeared most abundant, but what about at 5 or 7 dpi?
  2. Which dataset were used to make Figures 2 and 3?

The two figures show number of DEGs at 72 hpi and other time points, but the samples used for RNA extraction and sequencing were indeed collected at 12h, 24h, and 36h, respectively (Line 102).  

  1. It is necessary to verify the expression pattern/level of those potentially important DEGs using qPCR, which offer more convincing data for their findings, particularly for those genes reported in Lines 30-31.
  2. Lines 79-80 and Lines 231-234: The molecular mechanisms of pine tree interacting with Sirex noctilio venom may be rather complicated as it involves interaction between host plant and Sirex noctilio, actually three-part interaction between pine, Sirex noctilio and fungus in nature because Sirex noctilio uses two-component approach to damage pine trees and gets a source of nutrition for its developing larvae. It may be a good idea to conduct experiments to analyze molecular events in each of the three parts simultaneously, then integrative analysis would shed light on mechanisms for their interaction and host plant resistance. I noticed that you already published a similar work with Sirex noctilio recently, so you may relate the current findings with the previously published data in discussion.
  3. Lines 29-31: what are several new genes? What are various defense genes? They are the most important findings in this study as they may play important role in pine defense against Sirex noctilio venom; thus, they deserve more attention in this report rather than listed as supplementary (Table S3). Are the 21 DEGs listed in the Table equally important for host response to Sirex noctilio venom?

Some minor comments and suggestions are listed below:

Line 96:  Change ‘Nine healthy pines’ to nine healthy pine trees or pine plants. I assume they are susceptible genotype of Pinus 2 sylvestris var. mongolica, so is there any resistant genotype available?

Lines 96-98: Nine healthy pines…..were used in the experiment that involved three  treatments.

Are all nine pine trees the same genotype or at least derived from a single seedlot? Different plant genotypes may differentially respond the treatments.

Line 107:  What were the three biological replicates?

Line 418-424: conclusion needs re-writing to be more informative, so help the reader have a good idea of what the researcher has done and discovered.

Fig. 1 and 4 is difficult to read, so need to change font size to improve the readability.

Fig. 3 has very poor readability. In fact, this figure can be deleted as the simple information in the figure can be easily described in a sentence.

Author Response

This manuscript is a resubmission of an earlier submission. The following is a list of the peer review reports and author responses from that submission.